# Clinical Features of BoDV-1 Encephalitis: A Systematic Review

**Matteo Riccò** [1,*] , **Silvia Corrado** [2] , **Federico Marchesi** [3] **and Marco Bottazzoli** [4]

1 Occupational Health and Safety Service on the Workplace/Servizio di Prevenzione e Sicurezza Ambienti di Lavoro (SPSAL), Department of Public Health, AUSL–IRCCS di Reggio Emilia, 42122 Reggio Emilia, Italy
2 ASST Rhodense, Dipartimento Della Donna e Area Materno-Infantile, UOC Pediatria, 20024 Garbagnate Milanese, Italy; scorrado@asst-rhodense.it
3 Department of Medicine and Surgery, University of Parma, 43126 Parma, Italy; federico.marchesi@unipr.it
4 Department of Otorhinolaryngology, APSS Trento, 38122 Trento, Italy; marco.bottazzoli@apss.tn.it
* Correspondence: matteo.ricco@ausl.re.it or mricco2000@gmail.com; Tel.: +39-339-2994343 or +39-522-837587

**Simple Summary:** Fatal cases of encephalitis caused by Borna disease virus 1 (BoDV-1) have been increasingly reported. BoDV-1 causes animal Borna disease, an epidemic condition in pets and animals, with very high animal lethality, but very little is known about the clinical features of human encephalitis cases due to BoDV-1. The appropriate management of BoDV-1 cases requires a timely differential diagnosis from autoimmune encephalitis cases, whose treatment is based on the administration of immunosuppressive drugs that would be otherwise detrimental in viral encephalitis. Therefore, an up-to-date knowledge of BoDV-1 encephalitis clinical features is crucial for an appropriate and timely differential diagnosis. This review was meant to summarize all the available evidence on published cases of BoDV-1 encephalitis cases.

**Abstract:** Human cases of fatal encephalitis caused by Borna disease virus 1 (BoDV-1) have been increasingly reported. We envisaged the present systematic review in order to provide a comprehensive summary of clinical features associated with BoDV-1 encephalitis. Systematic research of four databases (PubMed, EMBASE, MedRxiv, BioRxiv) was performed during July 2023, and corresponding clinical and epidemiological data were collected and analyzed. A total of 37 BoDV-1 encephalitis cases from 15 different study cases and two countries (Germany, No. 35; France, No. 2) were detected, and their features were summarized (case fatality ratio, 91.9%). Age distribution followed a "U-shaped" distribution, with a first peak in individuals younger than 14 years (18.9%) and the second one in subjects older than 50 years (43.2%). Environmental risk factors were irregularly reported, but 17 out of 37 cases either lived in rural areas or reported repeated outdoor activities (45.9%). Interaction with pets and livestock was reported in eight cases (21.6%), stressing the zoonotic potential of BoDV-1 infections. Moreover, 16.2% of cases were reported among recipients of solid organ transplantations (five kidneys; one liver). Overall survival in children/adolescents vs. adults (≥18 years) was not significantly different (Hazard Ratio 0.878; 95% Confidence Interval from 0.366 to 2.105). Magnetic Resonance Imaging identified the involvement of basal ganglia, mostly of the caudate nucleus (42.4%) and thalamus (33.3%). Cerebrospinal fluid was often characterized by pleocytosis (78.4%). On the other hand, no distinctive clinical features were identified: initial symptoms were specific and included headache, fever, and confusion. In conclusion, BoDV-1 infection can result in fatal encephalitis, whose actual burden still remains unascertained. As the epidemiology of BoDV-1 is similarly elusive, encephalitis cases of unclear cause should be routinely tested for bornaviruses.

**Keywords:** BoDV-1; bornavirus; viral encephalitis; case reports; case series; systematic review

## 1. Introduction

*Bornaviridae* (order Mononegavirales) is a family of small (from 70 to 130 nm diameter for the enveloped particles, and from 50 to 60 nm for the viral core), negative sense, single-stranded, enveloped RNA viruses that infect a wide array of vertebrates [1–3], including

reptiles [4], birds [5], mammalians (horses, sheep, cattle, and rodents) [3,6], and even human beings [7–12]. To date, this family includes a total of 11 species, assigned to three genera [13], and three of those species have documented zoonotic potential: Borna Disease Virus 1 (BoDV-1); Borna Disease Virus 2 (BoDV-2); and the Variegated Squirrel Bornavirus 1 (VSBV-1) [2,10,12,14]. BoDV-1 has a small, highly conservative genome (8.9 kilobases) that is enclosed in viral particles with spherical geometry and helical capsid [10,12,15,16] and that encodes for at least six proteins: nucleoprotein (N, or p40); p10 (X); phosphoprotein (P, or p24); putative matrix protein (M, or gp18); type 1 membrane glycoprotein gp94 (G); and a viral polymerase (L, or p190) [17].

BoDV-1 infections in livestock (mostly horses and sheep) and domestic mammals can cause Borna disease (BoD) [15,18], a non-purulent meningo-myeloencephalitis character­ized by high case–fatality ratio, originally described in several countries of Central Europe (Germany, Liechtenstein, Switzerland and Austria) [2,14,19–23]. A putative host has been identified in *Crocidura leucodon* [19], the bicolored white-toother shrew, a small insecti­vore from the family of *Soricidae* [12,22,24,25]. Compared to other mammals, *C. leucodon* has shown a particularly high tolerance to BoDV-1 infection [19], being able to maintain the BoDV-1 infection at the local level and causing spillover events to pets and livestock through feces, urine, and saliva [19,26]. Since its original isolation in 1990, antibodies to BoDV-1 and viral RNA have been repetitively reported in human beings, suggesting the potential for human infections [15,27]. As high rates of seroprevalence for BoDV-1 were found in subjects characterized by autism, depression, and schizophrenia [15,28–30], a potential link between neurotropic BoDV-1 infections and chronic psychiatric conditions was suggested [29–33] and still remains highly debated [12,15,34,35]. For instance, despite earlier negative studies [36], a recent meta-analysis has suggested that BoDV-1 infections may be associated with schizophrenia [30].

Following the original description of human cases of VSBV-1 infections [37,38], the potential role of *Bornaviridae* as a potential cause of acute human encephalitis has been thereafter suspected [10,27,39–45], and a total of five cases from Germany [25,39,42] were described between 2018 and 2019. Interestingly, three of them occurred in recipients of solid organ transplants (kidney or liver) from an otherwise healthy donor who had died of alleged cardiac arrest [43,46], hindering a relatively high circulation of BoDV-1 as an indolent pathogen in the general population. The renewed interest in BoDV-1 encephalitis led to the retrospective analysis of 56 cases of encephalitis or encephalopathy of unknown etiology reported from the German state of Bavaria between 1995 and 2018 [12]. This post-mortem RT-qPCR study detected viral RNA in a total of eight cases, with further cases subsequently identified through active case finding [25]. By the end of 2022, a total of 35 cases of sporadic BoDV-1 encephalitis cases, all PCR-confirmed, have been notified to the German reference center (Robert Koch Institute) [25,27,39,43], and the number of reported cases is still increasing [44–47].

Despite the potential significance in clinical practice and the increasing availability of detailed clinical reports, features of BoDV-1 (meningo)encephalitis have not been sys­tematically reported. As a consequence, the present systematic review was designed to summarize and reconcile available data on clinical features of BoDV-1 encephalitis, as reported by published studies. Retrieved features were summarized in order to assess which clinical features may more properly help a timely differential diagnosis of BoDV-1, specifically focusing on cases occurring in children and adolescents (age < 18 years at the onset of symptoms) compared to adults (age ≥ 18 years at the onset of symptoms).

## 2. Materials and Methods

### 2.1. Study Selection, Inclusion, and Exclusion Criteria

The present study was designed according to the PRISMA statement (Prepared Items for Systematic Reviews and Meta-Analysis; see Supplementary File S1) [48,49] and then recorded in the PROSPERO (Prospective Register of Systematic Reviews) database with the ID number CRD42023454827.

Research concepts were defined according to the "PICO" strategy (Patient/Population/Problem; Investigated results; Control/Comparator; Outcome) as follows: in cases of BoDV-1 (meningo)encephalitis (P), the clinical features were reported in available studies (I, C), depending on the age at the onset of the clinical syndrome (O).

During July 2023, two scientific databases (i.e., PubMed and EMBASE) and the preprint repositories MedRxiv and BioRxiv were searched for entries on BoDV-1 encephalitis in humans without any chronological restriction, and the detailed research strategy is reported in Table A1. Moreover, a "snowball" approach was applied, with references to the retrieved studies being accurately searched for further suitable entries. For the purpose of this review, only the original research publications written in English, Italian, German, French, Spanish, and Farsi, including case reports and case series, were included in order to retrieve, where available, detailed clinical characteristics of individual cases of BoDV-1 encephalitis. Retrieved articles were initially assessed through title screening for their relevance to the subject [48,49]. Articles that were positively title-screened were then screened by the content of their abstracts. If that was considered consistent with the aims and the design of the present review, the full texts were independently assessed by two investigators (FM, SC) and abstracted.

In order to be considered consistent with this review and, therefore, included in the present systematic review, the following inclusion criteria had to be fulfilled:

(1) Availability of the full text;
(2) Diagnosis of viral encephalitis and or meningoencephalitis;
(3) Status of "probable" or "confirmed" BoDV-1 case according to RKI definition [25], that is: (a) Confirmed case: encephalitis or encephalopathy AND detection of BoDV-1 RNA in Cerebrospinal Fluid (CSF) or Central Nervous System (CNS) tissue, OR detection of BoDV-1 antigen by immunohistochemical analysis with virus-specific monoclonal antibodies in CNS tissue; (b) Probable case: Encephalitis or encephalopathy AND detection of bornavirus-reactive IgG in a serum or CSF sample by screening test (with full virus antigen, for example by immunofluorescence test), and suitable confirmation assay detecting antibodies against individual bornavirus antigens (derived from infected cells or recombinant antigens, e.g., the western blot, immunoblot, or ELISA).

### 2.2. Data Extraction

Data extracted included (where available) the following:

(a) Settings of the case: year, month or season, geographic region;
(b) Age and gender of the reported cases;
(c) Pre-existing clinical features, if any; in particular, previous solid organ transplantation(s), were taken into account;
(d) Potential risk factors: living in urban vs. rural areas; whether the patient worked as a farmer and/or with animal(s) and/or livestock; whether the patient(s) had any documented interaction with pets and/or livestock and/or rodents;
(e) Clinical characteristics at the onset of the symptoms. According to the clinical features reported by the original reports from Schlottau et al. [42], by Korn et al. [50], and by the case series of Niller et al. [12], the following signs and symptoms were specifically taken into account: flu-like syndrome (general aches and a fever); headache; fever (body temperature > 38 °C); apathy (loss of motivation, decreased initiative, and emotional blunting) [51]; asthenia; malaise, nausea and/or vomiting; any altered state of consciousness; any progressive loss of consciousness up to eventual coma; seizures; aphasia and/or blurred speech; hemiplegia or tetraplegia; sensorimotor neuropathy;
(f) CSF features at the onset of the symptoms: whether pleocytosis (>5 leucocytes/µL in CSF) [52] increased values of protein and/or lactate according to the normal range values of the parent institution;
(g) Features of electroencephalographic studies: whether focal or general anomalies were reported; signs of slowed rhythm;

(h)   Total T1/T2 anomalies reported at magnetic resonance imaging (MRI) studies at the onset of the clinical symptoms;

(i)   Outcomes: survival vs. death and weeks of total survival time.

If a certain patient was cross-posted by different studies, reports were accurately analyzed in order to fill the knowledge gaps, provide an extensive description of the clinical case, as well as to eliminate duplicates.

*2.3. Qualitative Assessment*

A qualitative assessment of retrieved studies was performed according to Murad et al. [53]. These authors proposed their instrument as specifically designed for case reports and case series. Each study is assessed in 4 domains (Selection, Ascertainment, Causality of case, and Reporting quality) through a total of 8 binary ("high risk" vs. "low risk") items. Two of them were specifically designed for studies on adverse drug events (D5, "Was there a challenge/rechallenge phenomenon?" and D6, "Was there a dose–response effect?"), all studies were rated according to the following items: D1: "Does the patient(s) represent(s) the whole experience of the investigator (center), or is the selection method unclear to the extent that other patients with similar presentation may not have been reported?" D2: "Was the exposure adequately ascertained?" D3: "Was the outcome adequately ascertained?" D4: "Were other alternative causes that may explain the observation ruled out?" D7: "Was follow-up long enough for outcomes to occur?" D8: "Is the case(s) described with sufficient details to allow other investigators to replicate the research or to allow practitioners to make inferences related to their own practice?".

All articles were rated according to the current indications by two investigators who independently read the full-text versions of eligible articles. Disagreements were resolved by consensus between the two reviewers; when it was not possible to reach consensus, input from a third investigator (M.R.) was searched and obtained. In accordance with the original recommendations from Murad et al. [53] and in analogy with the Risk of Bias (ROB) tool from the National Toxicology Program (NTP)'s Office of Health Assessment and Translation (OHAT) [54,55], even studies with "high" or "unclear risk" ratings in one or more of the assessed domains were included in the eventual body of evidence.

*2.4. Data Analysis*

The included studies were summarized by descriptive analysis. Crude prevalence figures per 100 people were therefore calculated. Their distribution was then assessed, referring to the variable of being aged <18 years vs. ≥18 years at the onset of clinical features. Corresponding distributions of categorical variables were initially analyzed through Fisher's exact test. Survival analysis was performed by calculation of Kaplan Meier survival curve.

Screening of retrieved articles was performed on Mendeley Reference Manager (version 2.97.0; Mendeley Ltd.; New York, NY, USA). All calculations were performed in R (version 4.3.0) [56], and RStudio (version 2023.03.0; RStudio, PBC; Boston, MA, USA) software by means of the packages "meta" (version 6.5-0), "fmsb" (version 0.7.5), "epiR" (version 2.0.63), and "robvis" (version 0.3.0). Plots were calculated by means of R packages "ggplot2" (version 3.4.3), "ggpubr" (version 0.6.0), "PRISMA2020" (version 1.1.1), and GraphPad Prism, Version 10.0 (GraphPad Software LLC, Boston, MA, USA).

## 3. Results

As shown in Figure 1, a total of 15 entries were ultimately retrieved [9,12,25,26,39,42, 46,47,50,57–62]; all of them were published after 2018. A total pool of 2869 entries (i.e., 99 from PubMed, 3.5%; 315 from EMBASE, 11.0%; 416 from MedRxiv, 14.5%; 2039 from BioRxiv, 71.1%) were initially identified. Of them, 706 (24.4%) were duplicated entries, being therefore removed (24.6%). The remaining 2163 articles were then screened by title and abstract; of them, 2130 were removed from the analyses as inconsistent with PICO (74.2% of the initial sample).

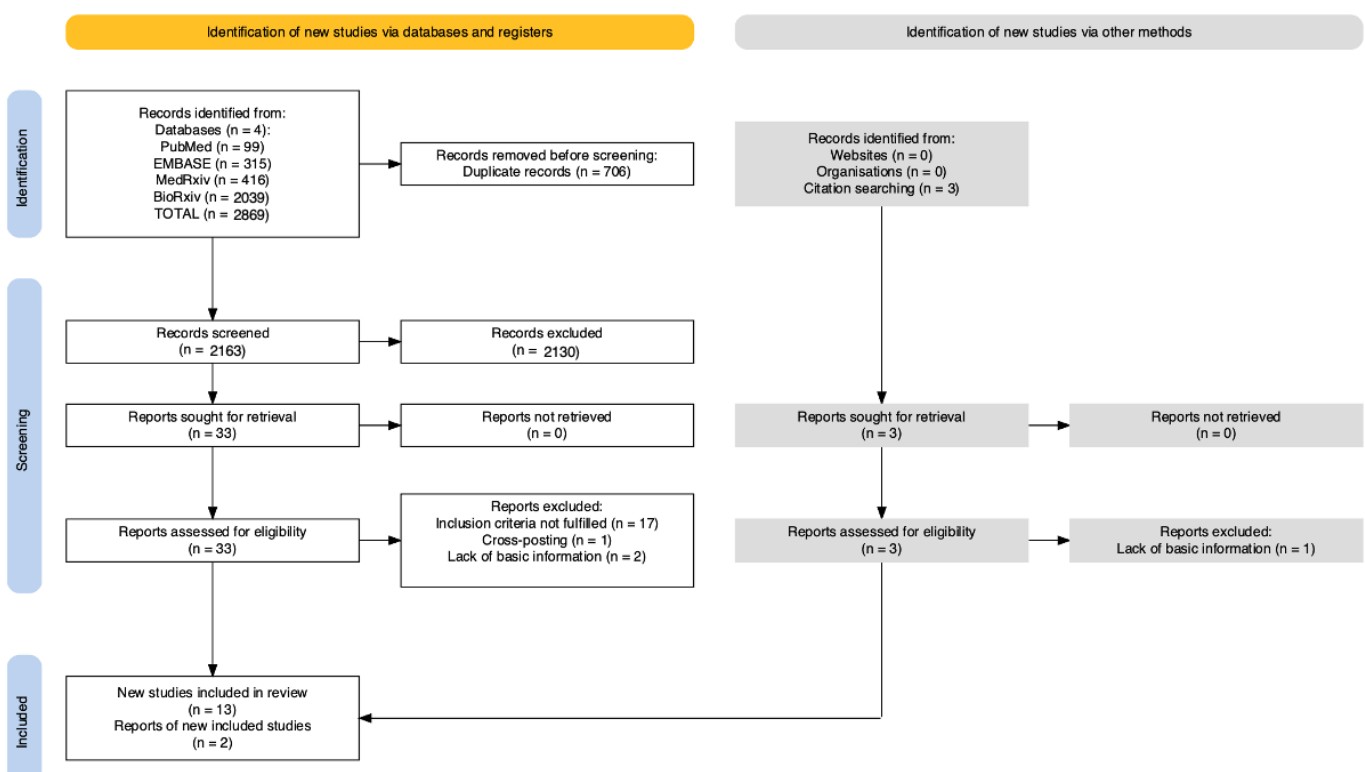

**Figure 1.** Flow chart of retrieved studies.

A total of 33 entries were assessed and then reviewed by full-text (1.2%); 17 of them were excluded as not fitting inclusion criteria (0.6%), while one article included duplicated reports, and two further reports were excluded as lacking basic information (i.e., outcome and length of the clinical syndrome since the onset of symptoms until discharge or death). The remaining 13 papers were eventually included in qualitative and quantitative analysis (0.5% of the initial sample), alongside 2 papers that were identified through analysis of references [46,61].

Overall (Table 1), five papers were individual case reports, and the remaining ones were case series, including 2 to 19 patients each. Indeed, with the notable exception of five reports [9,44,58,59,62], the large majority of the studies were characterized by some degree of cross-reporting, as one or more of the patients were otherwise included in subsequent studies. After the removal of duplicate cases, the eventual pool included a total of 37 cases reported since 1995.

**Table 1.** Summary of retrieved studies, and case reports included in the eventual summary.

| Study | Country | Reported Cases (No.) | Cross Reported Cases | Included Cases (No.) | Cases Included in the Pooled Analyses |
|---|---|---|---|---|---|
| Korn et al., 2018 [50] | Germany | 1 | Yes | 1 | P1 |
| Schlottau et al., 2018 [42] | Germany | 3 | Partially | 1 | P3 |
| Coras et al., 2019 [61] | Germany | 1 | Yes | 1 | P1 |
| Liesche et al., 2019 [47] | Germany | 6 | Partially | 5 | P1, P2, P3, P4, P5 |
| Finck et al., 2020 [60] | Germany | 19 | Partially | 11 | P1, P2, P3, P5, P6, P10, P12, P13, P15, P18; P19 |
| Niller et al., 2020 [12] | Germany | 8 | Partially | 5 | P1, P2, P3, P4, P5 |

**Table 1.** *Cont.*

| Study | Country | Reported Cases (No.) | Cross Reported Cases | Included Cases (No.) | Cases Included in the Pooled Analyses |
|---|---|---|---|---|---|
| Eisermann et al., 2021 [25] | Germany | 4 | Partially | 2 | P3, P4 |
| Schimmel et al., 2021 [58] | Germany | 2 | No | 2 | P1, P2 |
| Tappe et al., 2021 [38] | Germany | 1 | Yes | 1 | P1 |
| Bourgade et al., 2022 [62] | France | 2 | No | 2 | P1, P2 |
| Frank et al., 2022 [39] | Germany | 3 | Partially | 2 | P1, P2 |
| Liesche-Starnecker et al., 2022 [59] | Germany | 1 | No | 1 | P1 |
| Meier et al., 2022 [9] | Germany | 2 | No | 1 | P2 |
| Neumann et al., 2022 [46] | Germany | 1 | No | 1 | P1 |
| Grosse et al., 2023 [26] | Germany | 2 | Partially | 1 | P2 |

*3.1. Qualitative Assessment*

The overall quality of included studies is summarized in Figure 2. Briefly, the overall quality of included reports was mostly high or relatively high: with a cumulative score potentially ranging from 0 to 6, three studies (20.0%) scored 6/6, while six studies scored 5/6 (40.0%), and three studies scored 4/6 (20.0%). No study scored 3 or lower (Table A2).

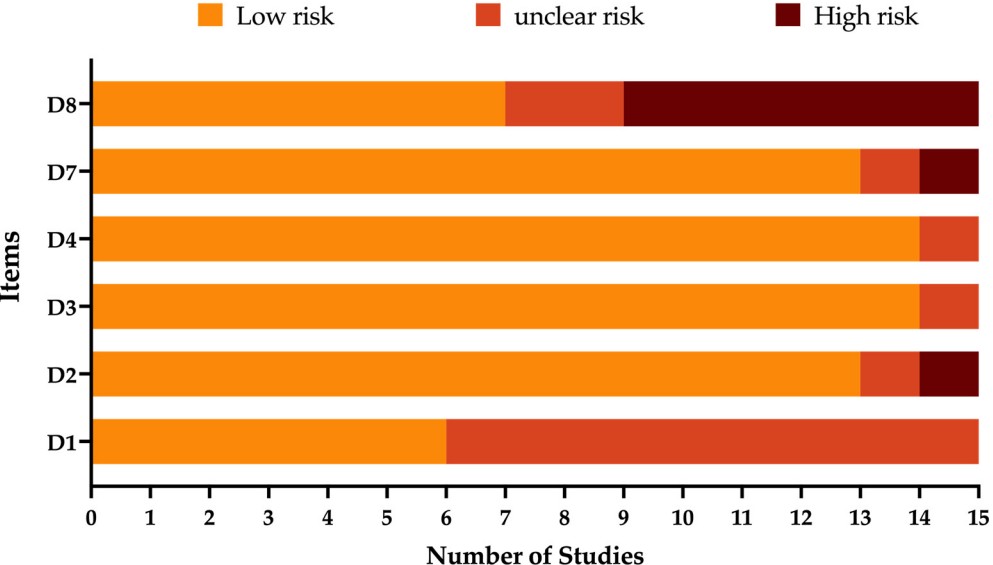

**Figure 2.** Quality assessment of 15 retrieved studies on Borna disease virus 1 encephalitis, according to Murad et al. [53]. Items D5 and D6 of the original model were removed as mostly relevant for the analysis of adverse drug events. Note: D1: "Does the patient(s) represent(s) the whole experience of the investigator (Center), or is the selection method unclear to the extent that other patients with similar presentation may not have been reported?" D2: "Was the exposure adequately ascertained?" D3: "Was the outcome adequately ascertained?" D4: "Were other alternative causes that may explain the observation ruled out?" D7: "Was follow-up long enough for outcomes to occur?" D8: "Is the case(s) described with sufficient details to allow other investigators to replicate the research or to allow practitioners to make inferences related to their own practice?".

The identification of pathogen (D2) and the eventual outcome of reported patients (D3) were well described in nearly all reports. Likewise, nearly all studies included BoDV-1 encephalitis as the final diagnosis after having accurately assessed other causes of viral encephalitis (D4). Moreover, all cases described the eventual outcome of the patients (D7). On the contrary, some concerns were identified when dealing with potential reporting bias (D1). On the one hand, most of the included patients [6,20,34,37,42,45] were representative

of the whole experience, either of the investigator or of the reporting center(s). On the other hand, the remaining studies did not accurately document the actual reporting strategy. Focusing on the accuracy of the reports, some concerns were raised by the studies by Bourgade et al. [62] and Schimmel et al. [58]. As the former was a research letter while the latter was a conference proceeding, the lack of details could be associated with the original design of the reports. Similarly, other papers [41,54] lacked clinical details because of their specific design, the former focusing on laboratory diagnostics and the latter on the pathological features of the reported case. Eventually, the otherwise high-quality studies from Finck et al. [60] and Niller et al. [12] lacked some details on the demographics of reported cases. Namely, these authors did not receive appropriate clearance from relatives of the described patients, while the studies of Frank et al. [39] and Eisermann et al. [25], despite a generally highly detailed report of clinical features, either lacked accurate reporting of potential risk factors, pre-existing clinical features, or laboratory and imaging studies from all patients.

### 3.2. Demographics

Available demographic data have been summarized in Table 2. Briefly, 35 out of 37 patients were reported in Germany (94.6%), most of them from the southern state of Bavaria (No. 32, 86.5% of total) [9,12,25,26,39,42,46,47,50,58–61]. One further case was reported from each of the three German states of Brandenburg [57], Saxony-Anhalt [39], and Thuringia [39]. The only two cases diagnosed outside Germany have been included in a single report from the Southern French region of Occitanie [62].

**Table 2.** Demographics of 37 patients with diagnosis of bornavirus encephalitis included in the analyses.

| Characteristic | No./37, % |
|---|---|
| Gender | |
| Male | 8, 21.6% |
| Female | 15, 40.5% |
| Not reported | 14, 37.8% |
| Age Group | |
| <20 years | 8, 21.6% |
| 20–50 years | 11, 29.7% |
| 50 years or more | 16, 43.2% |
| Not reported | 2, 5.4% |
| Region of origin | |
| Germany | |
| Bavaria | 32, 86.5% |
| Saxony-Anhalt | 1, 2.7% |
| Brandenburg | 1, 2.7% |
| Thuringia | 1, 2.7% |
| France | |
| Occitanie | 2, 5.4% |
| Reported environmental risk factors | 17, 45.9% |
| Residence in rural area | 9, 24.3% |
| Outskirts of urban centers | 5, 13.5% |
| Farming activities (any) | 5, 13.5% |
| Suburban activities (any) | 3, 8.1% |
| Any interaction with pets | 8, 21.6% |
| Any interaction with livestock | 2, 5.4% |

**Table 2.** *Cont.*

| Characteristic | No./37, % |
|---|---|
| Season (onset of symptoms) | |
| Winter | 4, 10.8% |
| Spring | 3, 8.1% |
| Summer | 8, 21.6% |
| Autumn | 2, 5.4% |
| Not reported | 20, 54.1% |
| Deaths | 34, 91.9% |
| Survival < 4 weeks | 6, 16.2% |
| Survival 4 to 9 weeks | 20, 54.1% |
| Survival ≥ 10 weeks | 4, 10.8% |
| Length of survival not reported | 4, 10.8% |

Overall, data on gender and age group were retrieved for 23 (62.2%) and 35 patients (94.6%), respectively. As explained by Niller et al. [12] and Bourgade et al. [62], more accurate reporting was impaired by ethical reasons. Nonetheless, 21 cases were females (40.5% of the total), while 8 cases (21.6%) occurred in individuals aged <20 years old, and six of them were aged 10 to 14 years old (16.2%) (Figure 3). The majority of cases occurred in adults, mostly over 50 years old (43.2%); among them, a total of eight cases (21.6% of the total) occurred in individuals aged 70 years or older. Even though the majority of cases did not report the actual timeframe associated with the onset of symptoms (54.1%), the higher share of cases occurred during the warm season (Summer months; 21.6%, Spring months: 8.1%).

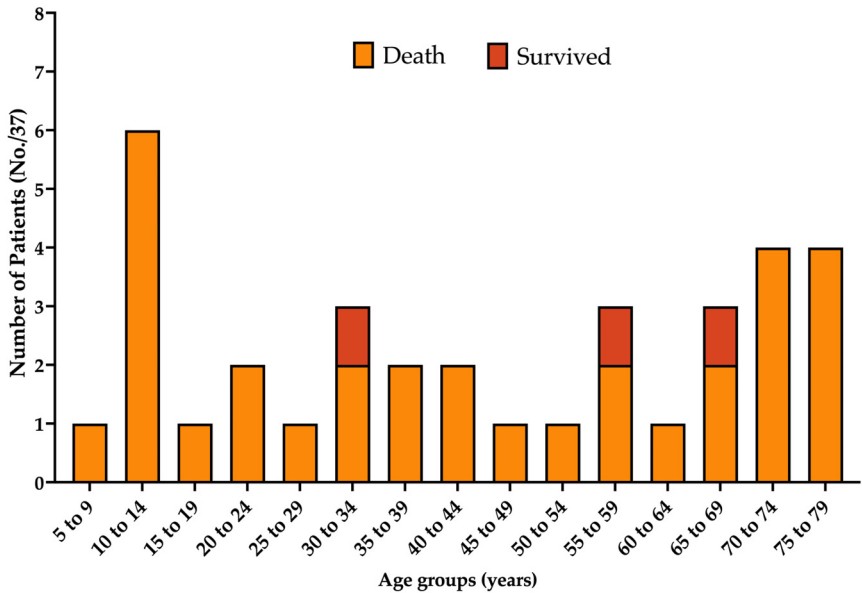

**Figure 3.** Distribution of 35 out 37 cases of encephalitis associated with Borna Disease Virus 1 (BoDV-1) by age group and eventual outcome.

Potential risk factors associated with the living environment were reported in only 17 out of 37 cases (45.9%). Overall, 9 out of 37 cases lived in rural areas (29.7%), while 5 further cases were from the outskirts of urban centers (13.5%). When dealing with potential occupational exposures, five cases (13.5%) were either professional or retired/hobby farmers. Potential interaction with pets was identified in eight cases (21.6%): 3 of the cases also reported previous interactions with rodents, and two had a cross-exposure to livestock, such as horses and/or cattle.

### 3.3. Pre-Existing Clinical Features

As shown in Table 3, pre-existing clinical features were inquired by 27 out of 37 cases (73.0%).

**Table 3.** Comorbidities of 27 out of 37 patients with diagnosis of bornavirus encephalitis included in the analyses.

| Comorbidities | No./27, % |
|---|---|
| Pre-existing comorbidities (any) | 8, 29.6% |
| Chronic Kidney disease | 5, 18.5% |
| Hepatic Cells carcinoma | 1, 3.7% |
| Diabetes | 3, 11.1% |
| Hypertension | 2, 7.4% |
| Multiple sclerosis | 1, 3.7% |
| Congestive heart disease | 1, 3.7% |
| Obesity | 1, 3.7% |
| History of solid organ transplantation | 6, 22.2% |
| Kidney | 5, 18.5% |
| Liver | 1, 3.7% |

The majority of the cases were described as otherwise healthy (19 out of 27 cases, 70.4%). Among pre-existing comorbidities, five cases were affected by chronic kidney disease (CKD), which in three cases was characterized as a complication of diabetes. All patients with CKD received kidney transplantation (18.5%) [12,42,47,62], with a further case of liver transplantation (3.7%) due to pre-existing hepatocarcinoma [42]. A further case was characterized by multiple sclerosis (3.7%), associated with congestive heart disease, hypertension, and obesity. According to available reports, only six patients with a previous history of solid organ transplantation were receiving immunosuppressive therapy at the time of the onset of BoDV-1 encephalitis.

### 3.4. Natural History

Overall, 34 out of 37 patients (91.9%) died because of BoDV-1 meningoencephalitis, with a median survival at four weeks after the onset of the clinical syndrome (range from 2 to 20). In four cases, while the eventual death was documented, the overall survival was not reported. Even among patients with documented survival, significant sequelae were reported, as the case reported by Coras et al. [61] required palliative care with mechanical respiration, the case published by Frank et al. [39] (P1) was managed in a nursing home at the time of the report, and the case reported by Schlottau et al. [42] in a patient having undergone liver transplantation (P3) was affected by optic nerve atrophy. Their detailed outcome is reported in Table A3. The majority of cases died because of complications of BoDB-1 between 4 and 9 weeks after the onset of symptoms (54.1%), with four cases reportedly surviving 10 weeks or longer (10.8%). The case fatality ratio in children and adolescents was estimated to be 100% compared to 92.6% of adults. However, no significant differences were identified between adults and children/adolescents (Hazard Ratio 0.878, 95% Confidence Interval 0.366 to 2.105; Gehan Breslow Wilcoxon test $p$-value = 0.602; Figure 4).

A likely spillover event was not identified in the large majority of reported cases, with the notable exception of three cases out of six of those who received solid organ transplantation [12,42]. Those three patients received an organ from the very same donor, a 70-year-old Bavarian male who did not have any previous neurologic signs or symptoms but whose grafts retained the BoDV-1 infection. The latency between the transplantation and the onset of clinical symptoms was 98 days for the single liver transplant and 80 and 112 days for the two kidney transplantations, respectively [42]. The longer latency estimates were reported for the two cases from the case series of Bourgade et al. [62] (9 years and 8 months), and the potential source of infection was not specifically inquired.

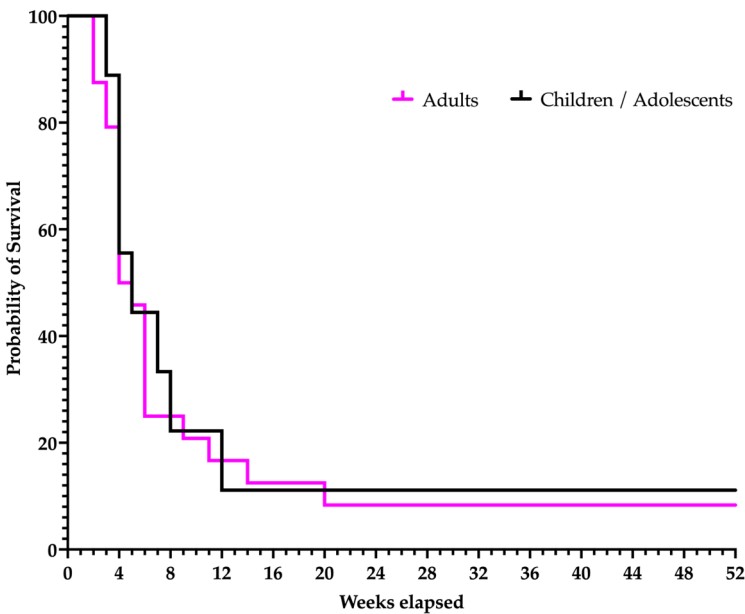

**Figure 4.** Survival analysis of children/adolescents (age < 18 y.o.; 8 cases, 87.5% case fatality ratio) compared to adults (age ≥ 18 y.o.; 27 cases; 92.6% case fatality ratio). No substantial differences were identified in children/adults compared to adults (Hazard Ratio 0.878, 95% Confidence Interval 0.366 to 2.105; Gehan Breslow Wilcoxon test *p*-value = 0.602).

### 3.5. Clinical Features

Clinical features of included cases have been reported in Table 4. In around one-third of the cases, the onset was defined as a "flu-like syndrome" (32.4%). However, the most common feature was represented by drowsiness (59.5%), followed by fever (56.8%) and headache (48.6%). Other non-specific symptoms were reported, such as nausea and vomiting (10.8%), malaise (10.8%), apathy (10.8%), asthenia (5.4%), arthralgia (2.7%), and even enuresis and weight loss (2.7% each).

Interestingly, 27.0% of the patients were characterized by a progressive loss of consciousness, and around 21.6% of the cases exhibited seizures within the first week of the symptom onset. A substantial share of cases was affected by impaired higher neurological functions, such as speech disturbances (21.6%, ranging from blurred speech to aphasia), memory deficits (10.8%), and even visual hallucinations (2.7%). Sensorimotor impairment was also commonly reported, as 27.0% of cases were affected by gait ataxia, with 10.8% of patients being affected by sensorimotor polyneuropathies. Moreover, three cases were either affected by hemiplegia (two cases, 5.4%) or tetraplegia (two cases, 5.4%).

As shown in Table 4, cases occurring in children and adolescents were characterized by higher rates of fever at onset (100% vs. 40.7%, *p* = 0.004) and gait ataxia (62.5% vs. 14.8%, *p* = 0.015). No other differences were identified in univariate analysis.

Focusing on cerebrospinal fluid features, the large majority of cases were characterized by pleocytosis (78.4%), while high levels of lactate and proteins were reported in 24.3% of total cases. No age-group differences were identified (all comparisons, *p*-value > 0.05).

Finally, EEG features were only reported in 14 cases (37.8%): 2 of them were negative (5.4%); in 10 cases, the rhythm was slowed (27.0%), and in 2 further cases general, non-focal anomalies were reported (5.4%). Because of the reduced number of samples, no univariate analysis by age group was performed.

**Table 4.** Clinical feature diagnosis of 37 patients with diagnosis of bornavirus encephalitis included in the analyses, in total and by age groups (age < 18 years vs. ≥18 years). In two cases, age at diagnosis was not provided, and corresponding reports were not included in the comparisons. Distribution of electroencephalographic (EEG) features was not performed, as reported from only 14 cases.

| Sign(s)/Symptom(s) | Total No./37, % | Age < 18 years (No./8, %) | Age ≥ 18 years (No./27, %) | Fisher's Test *p*-Value |
|---|---|---|---|---|
| Flu-like syndrome | 12, 32.4% | 2, 25.0% | 10, 37.0% | 0.685 |
| Headache | 18, 48.6% | 5, 50.0% | 12, 44.4% | 1.000 |
| Fever (at onset) | 21, 56.8% | 8, 100% | 11, 40.7% | 0.004 |
| Nausea/Vomiting | 4, 10.8% | 2, 25.0% | 2, 7.4% | 0.218 |
| Malaise | 4, 10.8% | 1, 12.5% | 3, 11.1% | 1.000 |
| Asthenia | 2, 5.4% | 0, - | 2, 7.4% | 1.000 |
| Apathy | 4, 10.8% | 0, - | 4, 14.8% | 0.553 |
| Drowsiness | 22, 59.5% | 4, 50.0% | 17, 63.0% | 0.685 |
| Progressive loss of consciousness | 10, 27.0% | 4, 50.0% | 4, 14.8% | 0.060 |
| Dysphagia | 3, 8.1% | 2, 25.0% | 1, 3.7% | 0.124 |
| Visual hallucinations | 1, 2.7% | 0, - | 1, 3.7% | 1.000 |
| Seizures | 8, 21.6% | 2, 25.0% | 5, 18.5% | 0.648 |
| Speech disturbances, including aphasia | 8, 21.6% | 1, 12.5% | 7, 25.9% | 0.648 |
| Hemiparesis | 2, 5.4% | 0, - | 2, 7.4% | 1.000 |
| Memory deficits | 4, 10.8% | 0, - | 4, 14.8% | 0.553 |
| Coma | 6, 16.2% | 0, - | 5, 18.5% | 0.315 |
| Meningism | 4, 10.8% | 1, 12.5% | 2, 7.4% | 0.553 |
| Polyneuropathy | 4, 10.8% | 0, - | 4, 14.8% | 0.553 |
| Gait Ataxia | 10, 27.0% | 5, 62.5% | 4, 14.8% | 0.015 |
| Enuresis | 1, 2.7% | 1, 12.5% | 0, - | 0.229 |
| Tetraplegia | 2, 5.4% | 0, - | 2, 7.4% | 1.000 |
| Weight loss | 1, 2.7% | 0, - | 1, 3.7% | 1.000 |
| Arthralgia | 1, 2.7% | 0, - | 1, 3.7% | 1.000 |
| Cerebrospinal fluid features | | | | |
| Pleocytosis (>5 leucocytes/µL) | 29, 78.4% | 8, 100% | 19, 70.4% | 0.154 |
| High lactate levels * | 9, 24.3% | 3, 37.5% | 4, 14.8% | 0.312 |
| High protein levels * | 9, 24.3% | 1, 12.5% | 6, 22.2% | 1.000 |
| EEG abnormalities | | | | |
| Slowed | 10, 27.0% | - | - | - |
| General abnormalities | 2, 5.4% | - | - | - |
| Negative | 2, 5.4% | - | - | - |
| Not reported | 23, 62.2% | - | - | - |

* according to the normal values of parent institution.

### 3.6. Imaging Features

MRI anomalies included any signal alteration reported by the parent study, and all documented features at the onset of the clinical syndrome are summarized in Table 5. In some cases, MRI findings were not reported, reducing the sample size to 33 cases (89.2%). As 19 of them were provided by the study of Finck et al. [60] as pooled estimates, no comparisons by the age group were possible, and only descriptive analysis is, therefore, provided.

Signs of CNS anomalies were documented in all brain areas, including the telencephalon, diencephalon, and brainstem, with a relatively high share of disorders affecting the diencephalon and the basal ganglia. In fact, 42.4% of cases were affected by anomalies of the head of the caudate nucleus, followed by the thalamus (30.3%). Also, the neocortex was frequently affected, with density anomalies reported in the temporal pole (30.3%) and insular cortex (30.3%), followed by the operculum, hippocampus, and parahippocampal gyrus (18.2% each). Interestingly, in three cases where the brainstem was affected, anomalies

were reported across all of its regions (i.e., mesencephalon, pons, and medulla oblongata; 9.1% each).

**Table 5.** Most frequently affected brain regions on Magnetic Resonance Imaging (MRI) at the time of the onset of clinical signs and symptoms.

| Region | No./33, % |
|---|---|
| Telencephalon | |
| Temporal pole | 10, 30.3% |
| Insular cortex | 10, 30.3% |
| Operculum | 6, 18.2% |
| Hippocampus | 6, 18.2% |
| Parahippocampal gyrus | 6, 18.2% |
| Gyrus rectus | 3, 9.1% |
| Occipital pole | 1, 3.0% |
| Deep white matter | 1, 3.0% |
| Optic nerves | 1, 3.0% |
| Diencephalon | |
| Head of the caudate nucleus | 14, 42.4% |
| Thalamus | 11, 33.3% |
| Putamen | 4, 12.1% |
| Brainstem | |
| Mesencephalon | 3, 9.1% |
| Pons | 3, 9.1% |
| Medulla oblongata | 3, 9.1% |
| Cerebellar hemisphere | 1, 3.0% |
| Pineal gland | 1, 3.0% |

## 4. Discussion

### 4.1. Summary of Main Findings

In this systematic review, data from 37 cases of BoDV-1 encephalitis were summarized. All cases were reported after 1995: the occurrence of a single case before 1995 was actually documented (more precisely, in 1992), but it was not included in the present summary because of the lack of inclusion criteria [39]. Interestingly, incidence and lethality both exhibited a "U-shaped" distribution, with two distinct peaks: the former in individuals younger than 14 years, while the latter in people older than 50 years. Even though no distinctive risk factors were ultimately identified, a large share of cases either lived in rural areas or reported repeated outdoor activities, including interaction with pets and livestock. Moreover, a relatively high share of cases (16.2%) occurred among recipients of solid organ transplantations, but in two cases, a very long latency was documented, suggesting that a different source of infection may be involved.

Acute viral encephalitis is far from being a rare occurrence. According to the Centers for Disease Control and Prevention (CDC), approximately 20,000 cases of acute viral encephalitis are reported annually in the United States alone, with a case fatality ratio (CFR) ranging between 5 and 20% [63], 7.3 encephalitis hospitalizations per 100,000 people from 2000 to 2010 [63,64], and with around half of incident cases that do not reach a definitive diagnosis [64–67]. Although rare, BoDV-1 encephalitis was characterized as a very severe condition. For instance, the overall CFR related we were able to assess (91.9%) is significantly higher than the one usually related to aseptic meningitis, as well as to acute (meningo)encephalitis cases caused by other viral pathogens (e.g., Herpes Simplex virus), as it usually does not exceed 20% [63]. Moreover, even patients surviving acute (meningo)encephalitis developed significant sequelae, including the requirement of mechanical ventilation [61], long-term care in a nursing home [39], and optic nerve atrophy [42].

Despite the relatively high quality of the parent reports, a substantial selection bias should be taken into account. In fact, our pooled sample included a substantial share of

cases that were only retrospectively retrieved, an approach that could have ultimately led to the oversampling of severe cases with dismal prognosis. A further possible bias may be related to the fact that a more aggressive clinical course can be related to a higher incidence of either post-mortem CNS biopsies or pre-mortem diagnostic assay, which could have led to a tardive, but more accurate diagnosis by RT-qPCR assays. Therefore, low-aggressive BoDV-1 infection may be underestimated. To date, the diagnosis of BoDV-1 infections remains particularly complicated by the lack of highly sensitive and specific commercial tests, while RT-qPCR testing has been only recently performed in the German states of Bavaria, Thuringia, Sachses-Anhalt, and Brandenburg, which are the areas of high animal BoD prevalence and where previous human cases of non-BoDV-1 infections occurred [25,45,68]. In this regard, the report of Bourgade et al. [62] is of particular interest. Following the publication of the case series of Schlottau et al. [42], Niller et al. [12], and Liesche et al. [47], the authors performed a retrospective assessment of a Southwestern France-based registry of kidney transplantation for cases of (meningo)encephalitis of unknown etiology, being able to identify at post-mortem RT-qPCR two further cases of BoDV-1. Overall, these findings further suggest that BoDV-1 may represent a substantial health threat for transplant medicine but also that the real-world occurrence of BoDV-1 infections could be far greater than otherwise hinted at by available reports and serosurveillance studies [12]. Moreover, even by acknowledging the solid organ transplantation as the source of BoDV-1 infection, a very long latency should be assumed, ranging between 8 months and 9 years. On the contrary, even though our current understanding of BoDV-1 infections in human beings does not allow us to rule out a very long latency period (as for Rabies virus, which also belongs to the order of Mononegavirales), with organs such as the liver and kidneys serving as reservoirs for the pathogen, this option eventually emerges as quite unlikely when we take in account the case series of Schlottau et al. [42]. In this case series, a total of three cases were reported among recipients of solid organ transplantations, and in all cases, the onset of the (meningo)encephalitis symptoms exceeded 4 months after the delivery of the graft.

Therefore, a key issue raised by our results is that the occurrence of BoDV-1-related (meningo)encephalitis cases in the general population is hard to reconcile with the known epidemiology of BoDV-1 infections in humans [10,14,24,43]. In fact, by summer 2023, a total of around 40 cases have been officially notified to RKI alone for the timeframe 1999–2023, and most of them have been included in the present review [9,10,43], with 2 further cases reported by Bourgade et al. [62]. Taking into account the likely reporting bias associated with the extensive referral to official registries, BoDV-1 (meningo)encephalitis could be acknowledged to be quite rare, yet far from being an unlikely event. Still, the implicit limits of collected reports impair our capability to ascertain the true burden of disease and its CFR. Even though earlier seroprevalence studies based on the sequential approach originally designed by Bode et al. hinted at very high prevalence rates [28,34,35,68–71], two recent and large studies from Germany have reported seroprevalence estimates for BoDV-1 that were <1%, even for areas characterized by active and documented circulation of the pathogen [10,43,72]. Again, these results urge for the definition of improved diagnostic strategies and more efficient diagnostic tests.

### 4.2. Summary of Clinical Features

Despite the overall features of human BoDV-1 encephalitis, we were able to describe a severe clinical condition, with progressive loss of consciousness and coma, with the eventual death of affected patients within 4 to 10 weeks since the onset of clinical symptoms, and no specific features could be reconciled with BoDV-1 infection. In most reported cases, the onset of BoDV-1 (meningo)encephalitis was associated with very common, unspecific features, such as an influenza-like illness, fever > 38°, and headache, a typical presentation for viral (meningo)encephalitis [63–67]. Cases of BoDV-1 encephalitis were then associated with a quite heterogeneous combination of signs and symptoms, including (but not limited to) convulsions, delirium, confusion, stupor or coma, aphasia or mutism, hemiparesis

with asymmetry of tendon reflexes and Babinski sign, involuntary movements, ataxia and myoclonic jerks, nystagmus, ocular palsies, and facial weakness.

Even though earlier reports suggested that BoDV-1 could exhibit features similar to the Guillaume–Barré Syndrome, with a rapidly progressing ascending tetraplegia [42,50,61], further reports rather dismissed this hypothesis, as only two cases of tetraplegia were documented. All of the features we were able to retrieve were quite commonly reported in cases of (meningo)encephalitis associated with other viral pathogens [63]. Moreover, our analyses suggested that onset in children and adolescents (age < 18 years) had a higher occurrence of fever (100% vs. 40.7%, $p = 0.004$) and gait ataxia (62.5% vs. 14.8%, $p = 0.015$) than in adults (age $\geq$ 18 years) at the onset of clinical symptoms. Even though the higher occurrence of fever is a consistent feature of viral infections in young children, a precautionary approach is requested, as both signs could reflect a later medical appraisal of included cases. On the one hand, flu-like syndromes in children and adolescents are quite common and could be, therefore, underscored until more specific features, such as gait disturbances, eventually appear. On the other hand, pediatric features of reported cases may reflect the greater attention of parents and familiars to uncommon anomalies.

*4.3. Summary of MRI Studies*

Analysis of MRI was more suggestive. According to available evidence, during the course of viral encephalitis, imaging studies of the brain are most often negative, with a limited occurrence of diffuse edema or enhancement of the cortex. While subcortical and deep nuclear involvement is rarely reported during the course of most cases of viral encephalitis [63], the involvement of the head of the caudate nucleus (42.4%), and the diffuse involvement of the thalamus (33.3%) is quite common in our paper. Unfortunately, because of the original design of the study from Finck et al. [60], which included around half of the MRI studies incorporated in this report, we were unable to calculate the cumulative occurrence of anomalies affecting basal ganglia or the whole of diencephalon, and the potential simultaneous occurrence of certain features as well. Interestingly, around one-third of our cases exhibited at the onset of some involvement of temporal lobes and insular cortex (30.3% each), followed by hippocampal and para-hippocampal areas, that collectively explain most of the reported signs and symptoms, and particularly speech disturbances, memory deficits, and, to some degree, even gait ataxia. These results are also quite consistent with the recent report from Grosse et al. [26], where the active replication of BoDV-1 was well-documented in the superior frontal gyrus, thalamus, amygdala, and superior/medial temporal gyrus, as well as in the olfactory bulb, that was, therefore, possibly identified among the potential portal of entry of BoDV-1 into the central nervous system.

*4.4. Summary of the CSF Studies*

As recently stressed in a case series from Neumann et al. [44], which summarized a total of 18 BoDV-1 human infections, 13 of them otherwise included in the present study, CSF changes in BoDV-1 encephalitis are comparable to those of other viral encephalitis. Even though the analysis of spinal fluid characteristics was commonly characterized by pleocytosis (72% vs. 78.4% from our estimates), the obvious corollary was that in 20 to 25% of cases, white blood cell count in CSF could be normal at the onset of clinical symptoms [44,60]. As T-cell pleocytosis is quite invariably reported in animal BoD, a likely explanation provided by Neumann et al. [44] is that CSF white blood cell count in BoDV-1 may show a certain dependency on the clinical course. However, it should be stressed that the analysis of CSF identified high content of protein and lactate in 24.3% of cases, possibly complicating the differential diagnosis of BoDV-1 (meningo)encephalitis in cases where pleocytosis is not reported, as the case would enter in differential diagnosis not only with other causes of aseptic meningitis or even with bacterial meningitidis [9,44,52,60].

On the other hand, the potential contribution of CSF analysis to the differential diagnosis of BoDV-1 cases is substantial. As recently summarized by Meier et al. [9], early

stages of BoDV-1 encephalitis substantially overlap with the clinical presentation of autoimmune encephalitis, with a subacute (generally less than 3 months) time frame and a wide spectrum of neuropsychiatric features ranging from progressive cognitive decline and symptoms such as seizures and focal neurologic deficits (e.g., aphasia, dysarthria, ataxia) [63]. While cases of BoDV-1 infection characterized by pleocytosis would enter the differential diagnosis with other viral infections, being, therefore, shortlisted for anti-viral treatment, cases without any sign of pleocytosis would be likely evaluated for a potential diagnosis of autoimmune encephalitis, whose appropriate treatment would require the delivery of immunomodulating drugs. As a consequence, CSF analysis of suspected BoDV-1 should be included in the diagnostic workup of patients with severe encephalitis of unknown cause [44].

### 4.5. BoDV-1 Encephalitis as a Zoonosis: Role of Rural and Occupational Exposures

A quite distinctive feature of BoDV-1 is the elusive identification of likely hosts and suitable vectors, including *Crocidura leucodon* [17,29]. However, the actual epidemiology of animal BoD only partially overlaps the distribution of bicolored white-toothed shrew [19,73,74], which includes central Europe eastwards to the Caspian Sea, the countries of the Balkan peninsula, Poland, Ukraine, countries from Caucasus area, Kazakhstan, and Iran [12,24,26], while animal BoD has been reported only in Central Europe, North America, and parts of Asia (Japan and Israel) [19,75]. Even though this apparent inconsistency might be only due to the lack of specific testing [75], a recent comparison of complete sequences and non-matching amino acid mutations of human isolates and shrews in the same cluster has stressed that the identification of *Crocidura leucodon* as the effective host for BoDV-1 may be quite problematic [34]. In turn, the alleged zoonotic nature of BoDV-1 has been, therefore, questioned. In fact, the data we collected, as well as the high prevalence of cases occurring in either rural/suburban settings and the reported exposures to pets, livestock, and farming activities, collectively confirm that human infections can result from occasional spillover events, particularly in agricultural settings and in individuals more likely to interact with animals (i.e., veterinarians, farm workers, etc.) [25–27]. These features have been previously stressed by Pörtner et al. [27], whose case-control study included 20 cases and 80 controls from their households. Still, even the aforementioned study was unable to identify a single plausible transmission event: the lack of suitable exposures in around half of the patients we included in our sample suggests that human-to-human transmission driven by otherwise healthy carriers should be considered [22,34], but the actual pathway remains elusive. Interestingly, in the recent study from Grosse et al. [26], very high numbers of viral copies were identified in the olfactory bulb of reported cases, and BoDV-1 transmission through the olfactory nerve route has been extensively proven in mammals such as horses and rodents. In other words, contaminated fomites and particles of dust would carry the pathogen within the nasal tract of airways, where BoDV-1 would find a suitable portal of entry in the olfactory mucosa [39,76–78]. Then, through neuronal transfer, viral copies would enter the olfactory bulb, whose diffuse connection with basal ganglia and neural cortex would explain the otherwise well-documented imaging and clinical features [2,3,73,76].

### 4.6. Limitations of Collected Results and Implication for Future Research

Our results are affected by substantial limits, including factors inherent in source studies and data collection. First of all, irrespective of their actual quality, the studies included in this data set were case reports and case series, and despite their relatively high quality, the total number of reported cases remains low [9,10,27,79]. Case reports and case series have been defined as descriptive studies that are prepared to illustrate novel, unusual, or atypical features identified in patients in medical practice [80]. To date, substantial disagreement remains on the value of case studies in the medical literature, particularly when dealing with the collection of evidence [9,10,27,53,79,80].

Second, as most cases were retrospectively retrieved from registries, with earlier episodes of BoDV-1 encephalitis occurring in the 1990s [47,59,60], the estimates for potential risk factors should be quite cautiously addressed, being reasonably affected by some degree of recall bias. As previously stressed, only 17 out of 37 cases provided some information about potential environmental sources of BoDV-1 infection, ranging from living in rural and suburban areas to occupational exposures (Table A4) [12,25,26,38,39,50,57,60]. However, as this specific topic was not systematically retrieved, we are unable to ascertain whether the remaining 20 cases lived in settings deprived of environmental risk factors or were simply not accurately retrieved for several reasons [44,46,47,58,60,62]. Nevertheless, it should be stressed that several studies have been affected by ethical reasons in their reporting accuracy, and the authors were forced to restrain the individual characteristics they were able to provide [12]. As a consequence, the actual association of individual risk factors with BoDV-1 infection should be very cautiously assessed, as otherwise suggested by a recent case-control study [27], and a precautionary approach is warranted. Similarly, current data do not allow for a conclusion about which comorbidities might increase the risk for BoDV-1 infection evolving to BoDV-1 (meningo)encephalitis because of their inconsistent reporting and potentially casual/spurious association with reported features, such as the kidney transplantation from the study of Bourgade et al. [62].

Third, even though most of the reported cases were from a limited geographic area (i.e., Southern Germany, Bavaria state), the present study includes patients from different medical centers [25,27,38,39,43,57,72,79,81]. Its multicenter nature may have resulted in heterogeneity of reporting due to clinical settings and interobserver variabilities. Because of the various allowances of care delivery and research resource availability across reporting hospitals, the present data set forcibly captures different diagnostic options, from clinical features to diagnostic tests and imaging options. These differences obviously limit the possibility of merging data across various sources. However, until more accurate studies are provided, pooling individual data from previously published case reports and case series will be the only available option for providing a comprehensive depiction of BoDV-1 encephalitis cases.

Fourth, the included patients were reasonably heterogeneous in terms of the clinical stage of their BoDV-1 infection [9,44,60,79]. In other words, reported clinical features may be reconciled with the clinical stage of BoDV-1 infection rather than the individual characteristics of that specific case. This is specifically relevant when dealing with MRI studies, as a common feature of viral encephalitis is the progressive involvement of several areas of the CNS, and that feature has also been previously recognized for BoDV-1 cases [60].

## 5. Conclusions

BoDV-1 infection is increasingly reported as a cause of severe (meningo)encephalitis cases. Even though the available evidence is limited, clinical features of BoDV-1 are quite similar to those of autoimmune (meningo)encephalitis, but the extensive involvement of the caudate nucleus and thalamus in MRI and the early report of pleocytosis from CSF studies could provide some contributions for an appropriate differential diagnosis. According to the available literature, BoDV-1 (meningo)encephalitis could be acknowledged as a quite invariably lethal disorder, but a substantial overestimation of its actual lethality cannot be ruled out. Collectively, our data stressed the importance of a more appropriate definition of the actual epidemiology of BoDV-1 infections.

**Supplementary Materials:** The following supporting information can be downloaded at https://www.mdpi.com/article/10.3390/zoonoticdis3040023/s1, File S1: PRISMA checklist. Reference [82] is cited in the Supplementary Materials.

**Author Contributions:** Conceptualization, M.R.; data curation, M.R. and M.B.; formal analysis, M.R. and F.M.; investigation, M.R.; methodology, M.B., F.M. and S.C.; resources, F.M.; software, M.R.; supervision, F.M. and M.B.; validation, S.C.; writing—review and editing, M.B. All authors have read and agreed to the published version of the manuscript.

**Funding:** This research received no external funding.

**Institutional Review Board Statement:** Not applicable.

**Informed Consent Statement:** Not applicable as all included data were retrieved from previously published studies.

**Data Availability Statement:** Data are available on request to the corresponding author.

**Conflicts of Interest:** The authors declare no conflict of interest.

**Appendix A**

**Table A1.** Research strategy and entries found by searched databases.

| Database | Keywords Searched | No. of Entries Found |
|---|---|---|
| PubMed [MeSH] | ("Borna Disease"[Mesh] OR "Borna disease virus"[Mesh] OR "Bornaviridae"[Mesh]) AND ("Encephalitis"[Mesh] OR "Encephalitis, Viral"[Mesh] OR "Encephalitis Viruses"[Mesh]) | 99 |
| EMBASE | ('orthobornavirus'/exp OR 'orthobornavirus' OR 'borna disease' OR 'borna disease virus' OR 'borna disease virus 1') AND ('encephalitis' OR 'brain disease' OR 'meningoencephalitis' OR 'meningitis' OR 'infectious meningitis' OR 'viral meningoencephalitis' OR 'virus meningitis') | 315 |
| MedRxiv | ("bornavirus" OR "borna disease virus") AND ("encephalitis" OR "meningitis" OR "meningoencephalitis") | 436 |
| BioRxiv | | 2039 |

**Table A2.** Summary of the methodological quality of included case reports and case series according to Murad et al. [53]. Items D5 and D6 were removed as mostly relevant for the analysis of adverse drug events. Note: ☺ = Low risk; ? = unclear risk; ☹ = High risk.

| Study | D1 | D2 | D3 | D4 | D7 | D8 | Score (0 to 6) |
|---|---|---|---|---|---|---|---|
| Korn et al., 2018 [50] | ☺ | ☺ | ☺ | ☺ | ☺ | ☺ | 6 |
| Schlottau et al., 2018 [42] | ☺ | ☺ | ☺ | ☺ | ☺ | ☺ | 6 |
| Coras et al., 2019 [61] | ? | ☹ | ☹ | ☺ | ☺ | ☺ | 3 |
| Liesche et al., 2019 [47] | ☺ | ☺ | ☺ | ☺ | ☺ | ☺ | 6 |
| Finck et al., 2020 [60] | ? | ☺ | ☺ | ☺ | ☺ | ? | 4 |
| Niller et al., 2020 [12] | ☺ | ☺ | ☺ | ☺ | ☺ | ? | 5 |
| Eisermann et al., 2021 [25] | ☺ | ☺ | ☺ | ☺ | ☺ | ☹ | 5 |
| Schimmel et al., 2021 [58] | ? | ☺ | ☺ | ☺ | ☺ | ☹ | 4 |
| Tappe et al., 2021 [38] | ? | ☺ | ☺ | ☺ | ☺ | ☺ | 5 |
| Bourgade et al., 2022 [62] | ? | ? | ☺ | ☹ | ? | ☹ | 1 |
| Frank et al., 2022 [39] | ☺ | ☺ | ☺ | ☺ | ☺ | ☹ | 5 |
| Liesche-Starnecker et al., 2022 [59] | ? | ☺ | ☺ | ☺ | ☺ | ☹ | 4 |
| Meier et al., 2022 [9] | ? | ☺ | ☺ | ☺ | ☺ | ☺ | 5 |
| Neumann et al., 2022 [46] | ? | ☺ | ☺ | ☺ | ☹ | ☹ | 3 |
| Grosse et al., 2023 [26] | ? | ☺ | ☺ | ☺ | ☺ | ☺ | 5 |

D1: "Does the patient(s) represent(s) the whole experience of the investigator (center), or is the selection method unclear to the extent that other patients with similar presentation may not have been reported? D2: "Was the exposure adequately ascertained?"; D3: "Was the outcome adequately ascertained?"; D4: "Were other alternative causes that may explain the observation ruled out?"; D7: "Was follow-up long enough for outcomes to occur?"; D8: "Is the case(s) described with sufficient details to allow other investigators to replicate the research or to allow practitioners to make inferences related to their own practice?".

**Table A3.** Detailed outcome of cases with documented survival.

| Case | Outcome |
|---|---|
| Coras et al. [61] | On day 5/6 after the onset of neurological symptoms, developed unresponsive state. After 4 months without any documented improvement, the patient was transferred to a local hospital closer to the family. After 6 weeks, the patient was released into the care of family in a palliative situation (mechanical respiration). |
| Frank et al. P1 [39] | The patient developed dysphasia, vigilance decline, and epileptic seizures, with sopor and ocular bulbus divergence. After 11 months, the patient was alive but severely disabled and lives in a nursing home. |
| Schlottau et al. P3 [42] | Developed symptoms associated with BoDV-1 infection 98 days after liver transplantation (facial palsy, anomia, cognitive deficits). After 9 months, the patient was alive with residual optic nerve atrophy. |

**Table A4.** Detailed reporting of reported environmental risk factors, comorbidities, and eventual outcome by original study and case identification.

| Reported Environmental Risk Factors (N. 17) | |
|---|---|
| Residence in rural area | Eisermann et al. P4 [25] <br> Frank et al. P1, P2 [39] <br> Korn et al. [50] <br> Liesche et al. P4 [47] <br> Niller et al. P3, P4, P5 [12] <br> Tappe et al. [57] |
| Outskirts of urban centers | Eisermann et al. P3 [25] <br> Grosse et al. P2 [26] <br> Niller et al. P1 [12] <br> Schimmel et al. P1, P2 [58] |
| Farming activities (any) | Korn et al. [50] <br> Liesche et al. P2, P4 [47] <br> Tappe et al. [57] <br> Finck et al. P15 [60] |
| Suburban activities (any) | Liesche et al. P4 [47] <br> Niller et al. P1, P2 [12] |
| Any interaction with pets | Liesche et al. P7 [47] <br> Tappe et al. [57] |
| Any interaction with livestock | Korn et al. [50] <br> Liesche et al. P7 [47] <br> Meier et al. P2 [9] <br> Niller et al. P1, P2, P3, P5 [12] <br> Tappe et al. [57] |

**Table A4.** *Cont.*

| Pre-existing comorbidities (N. 27) | |
|---|---|
| Chronic kidney disease | Bourgade et al. P1, P2 [62]<br>Liesche et al. P1 [47]<br>Niller et al. P3, P4 [12] |
| Hepatic cell Carcinoma | Schlottau et al. P3 [42] |
| Diabetes | Liesche et al. P8 [47]<br>Niller et al. P3, P4 [12] |
| Hypertension | Finck et al. P15 [60]<br>Niller et al. P3 [12] |
| Multiple sclerosis | Finck et al. P15 [60] |
| Congestive heart disease | Niller et al. P3 [12] |
| Obesity | Finck et al. P15 [60] |
| History of solid organ transplantation | Bourgade et al. P1, P2 [62]<br>Liesche et al. P1 [47]<br>Niller et al. P3, P4 [12]<br>Schlottau et al. P3 [42] |
| Kidney | Bourgade et al. P1, P2 [62]<br>Liesche et al. P1 [47]<br>Niller et al. P3, P4 [12] |
| Liver | Schlottau et al. P3 [42] |
| **Documented Survival (N. 3)** | Coras et al. [61]<br>Frank et al. P1 [39]<br>Schlottau et al. P3 [42] |

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
