# Peer review of "Clinical Features of BoDV-1 Encephalitis: A Systematic Review"

_zoonoticdis, doi:10.3390/zoonoticdis3040023_

Round 1

Reviewer 1 Report

Riccò et al. summarized current data on human cases on BoDV-1-induced encephalitis. The aim of the review is to collect all data availbale with regard to the patient profile such as age, gender, co-morbidities (to predict patients at risk) and to clinical features and total survival time (to discriminate BoDV-infection from autoimmune encephalitis as this is important for the therapy). Next, the authors dicussed the limitiation of an analysis of just 37 BoDV-1 cases. They assume that until now may be only the severe cases have been identified for BoDV and thus the total number of cases might be much higher. Thus, they hope that they can sensitize the clinic for checking for BoDV-1 in unclear mild cases of viral encephalitis.

In general, the overview is well done and data are sufficiently described, e.g. the quality criteria for including a case into the data analysis (Fig. 1), information that not all studies fullfilled each of the author´s questionary and that some of the patients were cross-posted by different studies. Nevertheless, in a few cases, data might be described more clearly and my proposals for changes are given in "major points".

Major points:

In the tables, some features might be described in another style for a better understanding. For example Table 2: data below Occitanie (page 8): Residence in rural areas /outskirts of urban centers: it is not clear why the total number is not 37, are the other cases reside somewhere else? and where? urban centers? or not reported?; farming activities/suburban activities/interaction with pets/livestocks, it is not clear whether and which cases are listed twice, e.g. the same patient which is doing farming activities might have also interactions with livestocks,

a possibility to overcome this problem (and to inform the reader on the double-listed cases) is to number each case (P1-P37) and give the percentage (e.g. interaction with pets = 8.1%) together with the patient ID (e.g. P7, P14, P23)

Figure 3: for the 3 patients which survived, it would be of interest to know the time period gone since they were diagnosed for BoDV or how long theses patients were without symptoms (just more than 10 weeks or even longer?), the reader may want to know whether these 3 patients really survived or whether they may die later

Table 3: I propose that information on co-morbidities such as diabetes which occur mainly or only in older people might be listed only for the 16 cases of > 50 years of age, (27 cases are also not cases >20 years of age, that would be 29 cases?, I guess the patients with unknown age were also not included), in addition the same patients with chronic kidney disease and liver cancer are found in the same table for solid organ transplantation, in my opinion these 6 cases (even getting immunosuppressive therapy) doesn´t help finding out which comorbidities increase the risk of a BoDV-infection, they might be not listed in this table and the information can be just mentioned in the text or presented in an additional Table, similarly 1 patient suffer of 3 listed comorbidities and thus there is only 1 patient with diabetes and hypertension: I recommend to state that the current data does not allow a conclusion which comorbidities might increase the risk for BoDV infection

Lines 321-322: may be I misunderstood, but according to Figure 3 the 3 surviving patients are 30-34, 55-59, and 65-69 years old, isn´t then the fatal ratio of children and adolescents 100%?

Table 4: the authors may shortly explain the meaning of the high lactate levels in the CSF, is this an indication of meningitis in general (just inflammation) or virus/or bacteria-induced meningitis

Table 4 on page 12: same proposal for another style (patient IDs) as for Table 2 since the same patients might have MRI anomalies in more than one brain region

Discussion: the authors may comment on the fact taht the virus can enter also other organs except of the brain, and may discuss whether in the other organs no symptoms occur, but they may serve as a reservoir (kidney, liver), line 542: the authors may add the literature speculations of the routes of entry of BoDV-1 (nose, intestine, lung?) - if available

Minor points:

Writing errors:

Line 17: …..encephalitis could be detrimental. Delete “could be detrimental”

Line 35 Cerebrospinal fluid

Line 391: BoDV-1

Line 401: letalithy

Line 422: may be

Line 519: autoimmune (to keep it homogenous thoughout the text)

Line 51: the authors mentioned 6 proteins, but later only 5 proteins were listed

Table 1: authors may define for the column Design the meaning of CR and CS (case report?), CR and CS both were used even when the report was only on 1 case

Table number 4 is given twice (pages 11 and 12)

Author Response

Estimated Rev.1,

first of all, we warmly thank you for your accurate and collaborative review, whose content would improve the overall quality of our study. Before replying point-to-point to your recommendations, please note that we did our best in order to implement the whole of your suggestion - only some issue regarding MRI studies have not been implemented for reasons that will be explained in the following lines.

More precisely:

1) In the tables, some features might be described in another style for a better understanding. For example Table 2: data below Occitanie (page 8): Residence in rural areas /outskirts of urban centers: it is not clear why the total number is not 37, are the other cases reside somewhere else? and where? urban centers? or not reported?; farming activities/suburban activities/interaction with pets/livestocks, it is not clear whether and which cases are listed twice, e.g. the same patient which is doing farming activities might have also interactions with livestocks...

a possibility to overcome this problem (and to inform the reader on the double-listed cases) is to number each case (P1-P37) and give the percentage (e.g. interaction with pets = 8.1%) together with the patient ID (e.g. P7, P14, P23)

REPLY:

Thank you for your suggestion. In fact (see following points) we revised Table 2 and the main text in order to clarify the lack of information that led to the doubts you've correctly stressed, i.e. in only 17 out of 37 cases potential environmental sources of infection were inquired. Some statement was provided within discussion, but we agree that is was not totally clear or transparent and it needed a more accurate description. Moreover, in order to provide a better insight on duplicated cases and multiple exposures, a new Appendix Table A4 was specifically designed. 

2) Figure 3: for the 3 patients which survived, it would be of interest to know the time period gone since they were diagnosed for BoDV or how long theses patients were without symptoms (just more than 10 weeks or even longer?), the reader may want to know whether these 3 patients really survived or whether they may die later.

REPLY:

thank you for your note. Not only main text was revised and now includes the following statement ("In 4 cases, while the eventual death was documented, the overall survival was not reported. Even among patients with documented survival, significant sequelae were reported, as the case reported by Coras et al. [61] required palliative care with mechanical respiration, the case published by Frank et al. [39] (P1) was managed in a nursing home at the time of the report, and the case reported by Schlottau et al. [42] in a patient having undergone liver transplantation (P3) was affected by optic nerve atrophy. Their detailed outcome is reported in Appendix Table A3") but a specifically designed Table A3 was specifically designed.

3) 

Table 3: I propose that information on co-morbidities such as diabetes which occur mainly or only in older people might be listed only for the 16 cases of > 50 years of age, (27 cases are also not cases >20 years of age, that would be 29 cases?, I guess the patients with unknown age were also not included), in addition the same patients with chronic kidney disease and liver cancer are found in the same table for solid organ transplantation, in my opinion these 6 cases (even getting immunosuppressive therapy) doesn´t help finding out which comorbidities increase the risk of a BoDV-infection, they might be not listed in this table and the information can be just mentioned in the text or presented in an additional Table, similarly 1 patient suffer of 3 listed comorbidities and thus there is only 1 patient with diabetes and hypertension: I recommend to state that the current data does not allow a conclusion which comorbidities might increase the risk for BoDV infection.

REPLY:

we totally agreed with your conclusions ("...  recommend to state that the current data does not allow a conclusion which comorbidities might increase the risk for BoDV infection") and such statement was specifically included in the discussion (Similarly, current data does not allow a conclusion which comorbidities might increase the risk for BoDV-1 infection evolving to BoDV-1 (meningo)encephalitis, because of their inconsistent reporting and potentially casual/spurious association with reported features, such as the kidney transplantation from the study of Bourgade et al.[62].)

However, we retained data for the whole of patients as we think that the aim and main significance of this study is providing of summary of available data, and that therefore we should share with potential readers all available data.

4) Lines 321-322: may be I misunderstood, but according to Figure 3 the 3 surviving patients are 30-34, 55-59, and 65-69 years old, isn´t then the fatal ratio of children and adolescents 100%?

REPLY:

please excuse us; we did some mistakes while updating the various proofs of the paper and it led to the improper reporting, that was then fixed (yes, CFR in children/adolescents was 100%)

5) Table 4: the authors may shortly explain the meaning of the high lactate levels in the CSF, is this an indication of meningitis in general (just inflammation) or virus/or bacteria-induced meningitis

REPLY:

thank you for your note; in fact we agree that the requested amendment (that was included in the discussion section on the CSF) improved the flow of the main text as follows:

However, it should be stressed that the analysis of CSF identified high content of protein and lactate in 24.3% of cases, possibly complicating the differential diagnosis of BoDV-1 (meningo)encephalitis in cases where pleocytosis is not reported, as the case would enter in differential diagnosis not only with other causes of aseptic meningitis or even with bacterial meningitidis [9,44,52,60].

6) Table 4 on page 12: same proposal for another style (patient IDs) as for Table 2 since the same patients might have MRI anomalies in more than one brain region.

REPLY:

unfortunately, this is the only amendment you've suggested we're unable to provide. In fact, as explained in the main text, "Unfortunately, because of the original design of the study from Finck et al. [60], that included around half of the MRI studies included in this report, we were unable to calculate the cumulative occurrence of anomalies affecting basal ganglia or the whole of diencephalon, and the potential simultaneous occurrence of certain features as well". 

7) Discussion: the authors may comment on the fact taht the virus can enter also other organs except of the brain, and may discuss whether in the other organs no symptoms occur, but they may serve as a reservoir (kidney, liver), line 542: the authors may add the literature speculations of the routes of entry of BoDV-1 (nose, intestine, lung?) - if available.

REPLY:

as this topic is hotly debated in the international literature, and unfortunately our data are inappropriate for providing any further insight, we implemented the following statements:

a) On the contrary, even though our current understanding of BoDV-1 infections in human beings does not allow to rule out a very long latency period (as for Rabies virus, that also belongs to the order of Mononegavirales), with organs such as liver and kidneys serving as reservoirs for the pathogen, this options eventually emerges as quite unlikely when we take in account the case series of Schlottau et al [42]. In this case series, a total of three cases were reported among recipients of solid organ transplantations, and in all cases the onset of the (meningo)encephalitis symptoms did exceed 4 months after the delivery of the graft

b) Interestingly, in the recent study from Grosse et al. [26], very high numbers of viral copies were identified in the olfactory bulb of reported cases, and BoDV-1 transmission through olfactory nerve route has been extensively proven in mammals such as horses and rodents. In other words, contaminated fomites and particles of dust would carry the pathogen within the nasal tract of airways, where BoDV-1 would find a suitable portal of entry in the olfactory mucosa [39,76–78]. Then, through neuronal transfer, viral copies would enter the olfactory bulb, whose diffuse connection with basal ganglia and neural cortex would explain the otherwise well documented imaging and clinical features [2,3,73,76].

Regarding minor points:

Line 51: the authors mentioned 6 proteins, but later only 5 proteins were listed

-> please excuse us, we "jumped" Protein X during the final setup of the main text.

Table 1: authors may define for the column Design the meaning of CR and CS (case report?), CR and CS both were used even when the report was only on 1 case

-> during this round of revision, we found that this information may be quite confusing as the table does include the total number of cases reported by each paper, so we removed that column

Table number 4 is given twice (pages 11 and 12) --> thank you, it was fixed.

Reviewer 2 Report

In this systemic review the authors, Ricco et al. did a very good job to review the clinical feature of BoDV-1 in human encephalitis. They also did an extensive review of the actual cases of rare but deadly BoDV-1 infection. This review is written in a very good way and it is very easy to understand which is a plus point as this review is important to the common people other than academicians or clinicians. The specific comments for this review are as below.

Major Comments:

1. The authors should include a paragraph for the pathogenesis of BoDV-1 infection in humans leading to encephalitis. Comparing the similarities and differences between the pathogenesis in animal and human would add more importance to this review.

2. The authors should check the result Line: 205-219 and the corresponding figure 1 to re-calculate the entries they have selected for the study. Specifically, in Line: 209-210 they removed 2133 articles from 2163 articles which results in 30 articles but the authors calculated as a total 33 articles were selected in Line:212. They should justify the extra 3 entries. Also, the authors should use the space in the boxes of figure 1 to make the text font larger to improve the clarity of the text.

Minor comments:

1. Line: 14-17; "Even though..... detrimental": Authors should re-write this complex sentence in simple sentences to make it more clear.

2. Line 24: Authors should add a timeline of the literature search from those databases.

3. Line 46: Authors should give corresponding references adjacent to the animal names for better correlation between text and references.

4. Line 301, 341, 363, 378 and 385: The table numbers are wrong and need to be changed.

5. Line 401, 422, 426: Spell correctly; lethality, may be, underestimated.

6. Line 419-420 and 446-448: The first line of both paragraphs need to be re-written as they are incomplete or unclear.

7. Line 455: Define 'CFR'.

8. Line 501, 545: Add punctuation; stop.

9. Line 516-520: Re-write 'While cases .... drugs' for clear understanding.

10. Line 565, 576: Authors start the sentences with 'Third' and 'Fourth'. They mentioned 'First of all in Line 548. They should mention second point in the revised manuscript.

Author Response

Estimated Rev. 2,

first of all, thank you for your collaborative and accurate review, whose content has reasonably contributed to the significant improvement of the quality of the present paper.

More precisely:

1. The authors should include a paragraph for the pathogenesis of BoDV-1 infection in humans leading to encephalitis. Comparing the similarities and differences between the pathogenesis in animal and human would add more importance to this review.

REPLY: thank you, we did it as follows: 

Interestingly, in the recent study from Grosse et al. [26], very high numbers of viral copies were identified in the olfactory bulb of reported cases, and BoDV-1 transmission through olfactory nerve route has been extensively proven in mammals such as horses and rodents. In other words, contaminated fomites and particles of dust would carry the pathogen within the nasal tract of airways, where BoDV-1 would find a suitable portal of entry in the olfactory mucosa [39,76–78]. Then, through neuronal transfer, viral copies would enter the olfactory bulb, whose diffuse connection with basal ganglia and neural cortex would explain the otherwise well documented imaging and clinical features [2,3,73,76].

2. The authors should check the result Line: 205-219 and the corresponding figure 1 to re-calculate the entries they have selected for the study. Specifically, in Line: 209-210 they removed 2133 articles from 2163 articles which results in 30 articles but the authors calculated as a total 33 articles were selected in Line:212. They should justify the extra 3 entries. Also, the authors should use the space in the boxes of figure 1 to make the text font larger to improve the clarity of the text.

REPLY: Thank you. In fact, we did a mess with the numbers because of the program we used for calculating Figure 1 (more precisely, the 3 papers that were initially retrieved from citation searching and snowball approach were improperly removed from the 33 retained in the pooled sources... please accept our apologies). However, both figure and main text were double checked and fixed. Now the content of the main text is as follows:

As shown in Figure 1, a total of 15 entries were ultimately retrieved [9,12,25,26,39,42,46,47,50,57–62], all of them published after 2018. A total pool of 2869 entries (i.e. 99 from PubMed, 3.5%; 315 from EMBASE, 11.0%; 416 from MedRxiv, 14.5%; 2039 from BioRxiv, 71.1%) were initially identified. Of them, 706 (24.4%) were duplicated entries, being therefore removed (24.6%). Remaining 2163 articles were then screened by title and abstract: of them, 2130 were removed from the analyses as inconsistent with PICO (74.2% of the initial sample).

A total of 33 entries were assessed and then reviewed by full-text (1.2%): 17 of them were excluded as not fitting inclusion criteria (0.6%), while 1 article included duplicated reports, and two further reports were excluded as lacking of basic information (i.e. outcome and length of the clinical syndrome since onset of symptoms until discharge or death). The remaining 13 papers were eventually included in qualitative and quantitative analysis (0.5% of the initial sample), alongside 2 papers that were identified through analysis of references [46,61].

Minor points

1. Line: 14-17; "Even though..... detrimental": Authors should re-write this complex sentence in simple sentences to make it more clear.

Thank you, the sentence was radically rewritten.

2. Line 24: Authors should add a timeline of the literature search from those databases.

Thank you, we specified in the main text the timeline of database searches and of retrieved documents.

3. Line 46: Authors should give corresponding references adjacent to the animal names for better correlation between text and references.

Thank you, the sentence was rewritten accordingly:

Bornaviridae (order Mononegavirales) is a family of small (70 to 130 nm diameter for the enveloped particles, and 50 to 60 nm for the viral core), negative sense, single stranded, enveloped RNA viruses that infect a wide array of vertebrates [1–3], including reptiles [4], birds [5], mammalians (horses, sheep, cattle, and rodents) [3,6], and even human beings [7–12].

4. Line 301, 341, 363, 378 and 385: The table numbers are wrong and need to be changed.

Thank you, we in fact decided during the final stage of development to implement Table 3 and 4 as distinctive ones, and therefore the numbers were totally messed up. It was fixed. 

5. Line 401, 422, 426: Spell correctly; lethality, may be, underestimated.

Thank you, fixed.

6. Line 419-420 and 446-448: The first line of both paragraphs need to be re-written as they are incomplete or unclear.

Thank you, we have reworked the main text.

7. Line 455: Define 'CFR'. Thank you, we define case fatality ratio.

8. Line 501, 545: Add punctuation; stop. Thank you, the sentences were rewritten.

9. Line 516-520: Re-write 'While cases .... drugs' for clear understanding.

Thank you, we extensively reworked that sentence for clearer understanding (While cases of BoDV-1 infection characterized by pleocytosis would enter in differential diagnosis with other viral infection, being therefore shortlisted for anti-viral treatment, cases without any sign of pleocytosis would be likely evaluated for a potential diagnosis of autoimmune encephalitis, whose appropriate treatment would require the delivery of immunomodulating drugs. As a consequence, CSF analysis of suspected BoDV-1 should be included in the diagnostic workup of patients with a severe encephalitis of unknown cause [44].)

10. Line 565, 576: Authors start the sentences with 'Third' and 'Fourth'. They mentioned 'First of all in Line 548. They should mention second point in the revised manuscript.

Thank you, we fixed the corresponding section.

Eventually, we would thank you again for your collaborative review and we're confident about the eventual acceptance of the revised text by Zoonotic Diseases.